# Physical and Dietary Intervention with *Opuntia ficus-indica* (Nopal) in Women with Obesity Improves Health Condition through Gut Microbiota Adjustment

**DOI:** 10.3390/nu14051008

**Published:** 2022-02-27

**Authors:** Karina Corona-Cervantes, Alicia Parra-Carriedo, Fernando Hernández-Quiroz, Noemí Martínez-Castro, Juan Manuel Vélez-Ixta, Diana Guajardo-López, Jaime García-Mena, César Hernández-Guerrero

**Affiliations:** 1Departamento de Genética y Biología Molecular, Cinvestav, Avenida IPN 2508, Ciudad de Mexico 07360, Mexico; karina.corona@cinvestav.mx (K.C.-C.); fernando.hernandez@cinvestav.mx (F.H.-Q.); juan.velez@cinvestav.mx (J.M.V.-I.); jgmena@cinvestav.mx (J.G.-M.); 2Departamento de Salud, Universidad Iberoamericana, Ciudad de México, Paseo de la Reforma 880, Ciudad de Mexico 01219, Mexico; alicia.parra@ibero.mx (A.P.-C.); noemi@outlook.com (N.M.-C.); dianagua_lo@outlook.com (D.G.-L.)

**Keywords:** fecal microbiota, obesity, tender cactus, nutrition, prickly pear

## Abstract

Obesity is a multifactorial disease resulting in excessive accumulation of fat. Worldwide, obesity is an important public health problem, affecting a large proportion of the world population. The tender cactus *Opuntia ficus-indica*, commonly known in Mexico as “nopal”, is widely distributed in this country, Latin America, South Africa, and the Mediterranean area. Nopal cladodes are commonly marketed in different forms as fresh, frozen, or pre-cooked, and used as fresh green vegetable. The aim of this study was to evaluate the capability of nopal to improve the health condition of participants affected by obesity, in a physical and dietary intervention, through gut microbiota modification. These results were contrasted with the effect of nopal in the gut microbiota of normal weight participants. We describe the association among biochemical, anthropometric markers, and the gut microbiota diversity found in fecal samples of the obese and normal weight groups. The results presented in this work suggest that caloric restriction, addition of nopal to the diet and physical activity, promote changes in the gut microbiota in obese women, improving the host metabolism, as suggested by the correlation between some bacterial species with biochemical and anthropometrical parameters.

## 1. Introduction

Obesity is defined as a multifactorial disease resulting in excessive accumulation of fat in adipose tissue, insulin resistance and chronic low-grade inflammation [1]. Obesity is measured using the body mass index; a person is considered obese if their BMI is higher than 30 kg/m^2^. It is one of the world’s major public health problems, affecting over 650 million people over 18 years of age [2,3].

Different factors have been associated with obesity: a high calorie diet, an increase in sedentary activity, and genetic factors. These causal factors could also be related to changes in the gut microbiota, resulting in disruption to the bacterial community, known as dysbiosis, affecting the host–microorganism relationship, energy homeostasis, metabolic function, and inducing inflammation [4,5]. There are several mechanisms involving gut microorganisms and the host, such as the regulation of fermentation and polysaccharide absorption from diet, protection of gastrointestinal mucosa permeability, etc. [3]. For this reason, there is a need for better intervention strategies to modify the gut microbiota, especially with diet modification, and addition of functional foods to improve the intestinal health of the obese population.

Nopal (*Opuntia ficus-indica*), commonly known as prickly pear, is widely distributed, mainly occurring in Mexico, Latin America, South Africa, and the Mediterranean area; its cladodes are marketed in different forms, such as fresh, frozen, or pre-cooked, and are used as a fresh green vegetable and salad [6]. Nopal has been used in traditional medicine due to its pharmacological properties, e.g., it acts as an anti-ulcerogenic, antidiarrheal antioxidant, anti-inflammatory, hypoglycemic, neuroprotective, and anti-hypercholesterolemic [6,7,8,9]. Some of these properties are attributed to the considerable number of different components in nopal, such as ascorbic acid, vitamin E, carotenoids, dietary fibre, amino acids, and antioxidant compounds (phenols, flavonoids, betaxanthin and betacyanin), providing health benefits [8].

In an excellent study made in a murine model, nopal diet intervention modified the gut microbiota and reduced the metabolic consequences of obesity, protecting from metabolic endotoxemia, and increasing the level of the protein occludin1 in the colon, reducing gut permeability [10]. Nevertheless, more studies are needed to evaluate the effect of nopal intervention in humans. The main aim of this study was to evaluate the capability of tender cactus, *Opuntia ficus-indica*, to improve the health condition of participants affected by obesity, in a physical and dietary intervention, through gut microbiota modification. These results were contrasted with the effect of nopal in the gut microbiota of normal weight participants. We describe the association among biochemical, anthropometric markers, and the gut microbiota diversity found in fecal samples.

## 2. Materials and Methods

### 2.1. Study Participants

A sample of 36 volunteer women were recruited among people attending the Nutrition Clinic at the Universidad Iberoamericana in Mexico City, between October 2018 to December 2019. The obesity group consisted of 25 women, and the normal weight group of 11 women. Women of the obesity group included participants who attended the Nutrition Clinic with the intention of receiving a typical nutritional intervention to lose weight and improve their life quality. The inclusion criteria were age of 18 to 59 years, BMI > 30 Kg/m^2^, no antibiotic treatment in the three months prior to the study and signing of the Privacy Notice of the Nutrition Clinic. The exclusion criteria were smoking, pregnancy, allergies, cancer, thyroid disease, atherosclerotic and cardiovascular disease, eating disorders, and consumption of any supplement in the last three months. Women of the normal weight group included participants who attended the Nutrition Clinic to monitor their health and nutritional status. For the normal weight group, the same inclusion and exclusion criteria described for the obesity group were applied, except that a BMI of 18.5–24.99 was required. All participants signed informed consent in accordance with the Helsinki Declaration, revised in 2013. The ethics and scientific committee of the Universidad Iberoamericana Mexico City approved the research (protocol ID No. 2018-01).

### 2.2. Anthropometric Measurements

Body weight and percentage of fat mass were measured by InBody Model 720 (0.1 kg accuracy, 250 kg capacity, Biospace Co., Seoul, Korea), under fasting conditions in all the participants. Height was measured using a stadiometer (Seca^®^ Model 240; 2 mm accuracy, Seca GmbH & Co., Hamburg, Germany) with participants in standing position after removal of shoes. The waist was measured at the midpoint between the lower rib and iliac crest.

### 2.3. Dietary Intervention

This was a prospective dietary intervention study. Each participant of the obesity group received a personalized diet plan with an energy restriction of 500 kcal per day, based on their predicted TEE (Total Energy Expenditure), with a nutrient distribution of 50% carbohydrates, 25% proteins 25% fats, and fiber intake of at least 25 g/day. The diet plan of each participant in the obesity group included 300 g (2 cups) of boiled nopal (*Opuntia ficus-indica*) obtained from 375 g of fresh nopal cladodes (100 g = 16 kcal and 2.2 g of dietary fiber), which corresponds to 3% of the total energy intake and 33% of the daily recommendation for dietary fiber (25 g) based on a 2000 kcal diet. Participants also received an indication to walk at a ‘brisk pace’ for 30 min per day.

To standardize the source, ripening, and preparation of nopals among participants, and to reduce barriers for participants accessing the indicated food portions, they received 2100 g of nopal in sterile plastic bags every week at no cost. Participants received nopals every week at the Nutrition Clinic of Universidad Iberoamericana. Fresh nopals were obtained by the university from same provider and production area. The nopals were prepared similarly to household practices; the spines were removed, the cladodes were chopped and cooked in boiling water for 11 min and placed on ice to cool. After boiling, the liquid was drained off, packed in sterile plastic bags, and refrigerated, before distribution. The normal weight group received the same nopal portions described above. This group received no personalized diet plan with energy restriction during the study and continued their normal lifestyle.

Participants received a manual to record their adherence to the nutritional intervention and the indication of their physical activity during their first visit to the Nutrition Clinic and consecutive weeks. Participants who reported less than 80% adherence to the supplementation in follow-up visits or did not finish the one-month intervention with energy restriction diet and nopals, were discarded from the study.

### 2.4. Ethical Considerations

This protocol was revised and approved by the Universidad Iberoamericana Ciudad de México scientific and ethical boards (ID# 2018-01). All women were invited to participate in the study and received information related to the study; they signed informed consent in accordance with the Helsinki Declaration revised in 2013.

### 2.5. Determination of Biochemical Variables in Peripheral Blood

Peripheral fasting blood samples were collected into 7 mL heparin tubes (BD Vacutainer, Franklin Lakes, NJ, USA), at the beginning and end of the dietary intervention (30 days). The concentration of glucose, total-cholesterol, LDL-cholesterol, HDL-cholesterol, and triglycerides was determined by Alere-Cholestec LDX System (Alere, San Diego, CA, USA). The assays were conducted according to the manufacturer’s instructions and recommendations. The intra-assay variation of all biochemical determinations described above was less than 5%, and the inter-assay variation was less than 10%. All clinical, biochemical, anthropometric, and laboratory tests were carried out by certified dietitians and/or trained staff.

### 2.6. Bacterial DNA Extraction from Stool Samples

Fecal samples were collected aseptically in a sterile stool container, and transported to the laboratory using ice packs, aliquoted and immediately stored at −70 °C until further processing. DNA was extracted from 0.15 g of feces using the ZR Fecal DNA MiniPrep (Zymo Research, Irvine, CA, USA). The quantity of purified DNA was measured by its 260/280 nm absorbance ratio using a NanoDrop Lite Spectrophotometer (Thermo Scientific, Waltham, MA, USA), and the quality was evaluated by electrophoretic fractionation in 0.5% agarose gels.

### 2.7. V3 16S rDNA Libraries Preparation

For DNA libraries preparation, an amplicon of approximately 281 bp, including the V3 polymorphic region of the bacterial 16S rRNA gene, was amplified using a sense V3-341F primer containing different 12 bp Golay barcodes for each sample [11], plus the A-adapter for massive sequencing in Ion Torrent PGM (Life Technologies, Carlsbad, CA, USA). Antisense V3-518R (Reverse) primer contained the P1-adapter. PCR was made in 50 μL reaction as previously described [12] using GeneAmp PCR System 2700 Thermocycler (Applied Biosystems, Waltham, MA, USA), and the program was 5 min 95 °C; 25-cycles [15 s, 94 °C; 15 s, 62 °C; 15 s, 72 °C], followed by an extension of 10 min at 72 °C.

### 2.8. Massive Semiconductor Sequencing

Purified pooled amplicons of fecal DNA were sequenced in-house as previously reported [11]. In brief, the size and amount of DNA fragments per micro liter were calculated using Agilent Bioanalyzer 2100 (Santa Clara, CA, USA), and libraries for each run were diluted to 26 pM prior to clonal amplification. Emulsion PCR was carried out using the Ion OneTouch^TM^ 200 Template Kit v2 DL (Life Technologies, Carlsbad, CA, USA) according to the manufacturer’s instructions. Amplicon enrichment with ion spheres was done using Ion One Touch^TM^ ES system (Waltham, MA, USA). The sequencing was done using Ion 318 v2 Chips and Ion Torrent PGM system (Waltham, MA, USA). After sequencing, reads were filtered by the PGM software to remove low quality and polyclonal sequences. During this process sequences matching the 3′-adapter and 5′-adapter were automatically trimmed and filtered.

### 2.9. Analysis of Sequenced Data for Microbial Diversity

Ion torrent PGM software, Torrent Suite v4.0.2 was used to demultiplex the sequenced data of the fecal microbiota, based on their barcodes. Poor quality reads were eliminated from the datasets, i.e., quality score ≤ 20, containing homopolymers > 6, length < 200 nt, and containing errors in primers or barcodes. Filtered data were exported as FASTQ files. Demultiplexed sequencing data were analyzed using QIIME 2 (v2021.4) pipeline [13]. FASTQ files were imported into artifact file demux.qza (fastq + manifest file with the sample-id and absolute-filepath of fastq files). Then the demux.qza was denoised using dada2 v2021.2.0.

### 2.10. Determination of Microbiota Relative Abundance, Diversity, and Significant Enrichment

Taxonomic classification was made using qiime feature-classifier classify-consensus-blast (v2021.2.0), Greengenes database (v13.8), qiime feature-table group, qiime metadata tabulate, and qiime taxa barplot. Alpha-diversity was characterized by Shannon, Simpson, Chao1 indexes and observed number of features using phyloseq (v1.22.3), and ggplot2 (v3.1.0) packages in R (v3.4.4) [14]. The beta diversity was made using qiime diversity beta-phylogenetic (v2021.2.0) which calculated the dissimilarity index using UniFrac distance metric as % of total variability in different axes of the plot and visualized by principal coordinate analysis (PCoA). Linear discriminant analysis (LDA) effect size (LEfSe) v1.0 was used to determine differences in the relative abundance of bacterial taxa between groups. The parameters for the analysis were *p* ≤ 0.05, LDA score 2.0 with default, and results graphically represented as bar plots [15].

### 2.11. Core Microbiota Determination and Heat Map

To determine the bacterial taxa, present in 95% of the faecal samples, the qiime feature-table core-features program from QIIME 2 (v2021.2.0) pipeline was used with the feature and taxonomy table files. The counts for each common bacterium were tabulated in TSV file format, and the natural logarithm determined for each data entry. A heat map of abundance was constructed using the gplots (v3.1.1) and RColorBrewer (v1.1–2) packages with script for R study program from GitHub website https://github.com/rasbt/R_snippets/blob/master/heatmaps/h2_default_clustering.R (accessed on 20 Febuary 2022).

### 2.12. Multivariate Association Analysis

To investigate the associations between bacterial taxa abundances and metadata, multivariate association with linear models (MaAsLin, v0.0.4) analysis was performed using default parameters in R (v3.4.4). This multivariate analysis generates significant association graphs reporting *p*- and *q*-values; *p*-values < 0.05 and *q*-values < 0.25 were considered statistically significant [16].

### 2.13. Predicted Metabolic Pathways of the Gut Microbiota

PICRUSt2 (Phylogenetic Investigation of Communities by Reconstruction of Unobserved States) (v2.4.1) [17] was used for the analysis. Closed reference OTUs were picked up with 97% similarities against Greengenes database (v13.8) in QIIME 2 (v2021.4), generating a feature-table.biom file. This file was used to predict the metabolic pathways of bacteria with the help of the Kyoto Encyclopedia of Genes and Genomes (KEGG) database. Statistical analysis of metagenomic profiles software STAMP (v2.1.3) was used with Welch’s *t* test [18] to visualize and analyze the significant values.

### 2.14. Spearman Correlation Analysis

Spearman correlation analysis was made with the bacterial taxa at level 6 and numerical clinical data text files using microbiome (v1.14.0) package. The correlated data set was filtered using dplyr (v1.0.7) to remove the data with *p*-value < 0.05, and finally ggplot2 (v3.3.5) was used to obtain the heat map with correlation values [19].

### 2.15. Statistical Analyses

Epidemiological, clinical, and biochemical variables were analyzed using descriptive statistics of both groups. A *t*-paired test for parametric data or Wilcoxon Signed Rank Test for non-parametric data was applied to assess the differences between the beginning and end of the supplementation. Differences with a *p*-value < 0.05 were considered statistically significant. Statistical analyses were done using SPSS (v24.0). Relative abundances and alpha diversity values were analyzed using one-way ANOVA, pairwise PERMANOVA with 999 permutations. Data were expressed in means ± SD and *p*-values < 0.05 were considered statistically significant. To correct the *p*-values, multiple testing Benjamini–Hochberg correction analysis [20] was done using the p.adjust() function of R (v3.4.4), to avoid the inclusion of false positives in LEfSe (v1.0). In addition, statistical analysis of metagenomic profiles software (STAMP, v2.1.3) was used to perform the statistical analysis of the PICRUSt2 (v2.4.1) data.

### 2.16. Sequence Accession Numbers

The sequence FASTQ files, and the corresponding mapping file for all samples used in this study, were deposited in the NCBI BioProject ID PRJNA783637 Link https://www.ncbi.nlm.nih.gov/bioproject/PRJNA783637 (accessed on 20 February 2022).

## 3. Results

### 3.1. Anthropometric and Biochemical Parameters Evaluated in the Obesity and Normal Weight Groups

The obesity group included 25 participants with obesity who underwent a calorie-restricted nutritional intervention, supplemented with a daily dose of 300 g boiled nopal for 30 days. The average BMI of this group at the beginning of the study was 35.1 ± 4.5, and it was 34.1 ± 4.7 by the end of the intervention (*p* < 0.005). Likewise, the participants also showed statistically significant decrease in weight, hip, waist/hip ratio, glucose, total cholesterol, and HDL-cholesterol (Table 1). On the other hand, participants in the normal weight group did not show any statistically difference with the use of the nopal supplementation during the study (Table 1). A statistically significant difference was found in the average age between groups: the obesity group had an average age of 40.6 ± 10.7, the normal weight group had an average age of 22.1 ± 2.6 (*p* < 0.05). This age difference is associated to the high prevalence of overweight and obesity, which increases to a value of 76%, in people of 20 years old and more. 

### 3.2. Composition of the Faecal Microbiota Is Not Associated with Nopal Diet Intervention

Fecal samples were collected from women with normal weight and women with obesity at the beginning and end of the intervention, DNA was extracted, V3-16S rRNA gene libraries were prepared by PCR, and high-throughput sequenced, as described in the Materials and Methods section. A sequencing set of average 183 bases, with at least 9150 counts, with a Phred-like Quality Score of 31 was obtained for each woman (Appendix A). At phyla level, the relative abundance of Bacteroidetes and Firmicutes were dominant in all groups; only the normal weight group at beginning of the intervention showed high values of relative abundance for Bacteroidetes, and less relative abundance of Firmicutes comparing with the rest of the groups; nevertheless, this difference is not statistically significant (Figure 1, Appendix A). Alternatively, the relative abundance of phylum Firmicutes was at least 50% and the Bacteroidetes was at least 30% in all groups. The phyla Proteobacteria and Actinobacteria were at least 1.5% in all groups, and Cyanobacteria had the lowest relative abundance or was not detectable in the obesity group by the end of nopal diet intervention. No statistically significant changes at phyla level were observed with the nopal intervention, neither in the normal weight or the obesity group (Figure 1, Appendix A).

The analysis at order, family and genus level showed that the families *Lachnospiraceae*, *Blautia* and *Prevotella* were present in all groups, being more represented in the obesity group, with a tendency to increase with the nopal intervention. On the other hand, *Bacteroides* and *Ruminococus* were more abundantly present in the normal weight group at the beginning of the intervention, in lower proportion in the obesity group, and with a tendency to decrease with the nopal intervention (Appendix A).

### 3.3. The Alpha and Beta Diversities of Faecal Microbiota Were Not Affected by the Intervention with Nopal

The alpha diversities of the fecal microbiota did not show statistically significant differences after nopal diet intervention (Figure 2, Appendix A).

Regarding beta diversity, pairwise PERMANOVA analysis using beta diversity data did not show any statistically significant difference between the beginning and end of the intervention for the normal weight group (*p* = 0.916, *q* = 0.916); neither for the beginning and end of the obesity group (*p* = 0.686, *q* = 0.823) (Appendix A).

### 3.4. Nopal and Caloric Restriction Diet Intervention Lead to an Increase of Specific Bacteria in Faecal Samples

For the obesity group, the core microbiota present in at least 95% of all fecal samples, showed a decrease in the absolute abundance of families like *Ruminococcaceae*, *Lanchnospiraceae*, and the genera *Prevotella* (*Prevotellaceae*), and *Ruminococcus* (*Lanchnospiraceae*), *Streptococcus* (*Streptococcaceae*), and *Bifidobacterium* (*Bifidobacteriaceae*). For the normal weight group, there is a decrease in the family *Barnesiellaceae*, and the genus *Bacteroides* (*Bacteroidaceae*) (Figure 3).

The LEfSe analysis that reveals taxa with larger relative abundance with statistical significance showed that the genus *Parvimonas* was increased at the beginning of the intervention with respect to the end of dietary intervention in the obesity group (Figure 4). The normal weight group showed no differences according to the LEfSe analysis.

### 3.5. Association Analysis of Clinical Metadata with Faecal Bacteria during Dietary Intervention with Nopal

We determined a statistically significant association of members of the gut microbiota with the clinical metadata using MaAsLin analysis. The results showed negative association of BMI with the abundance of the family *Barnesillaceae* (*p* = 0.0112, *q* = 0.194) for the obesity (29–45) and normal weight groups (19–24) (Figure 5).

### 3.6. Correlation Analysis of Clinical Metadata with Faecal Bacteria during Dietary Intervention with Nopal

We used Spearman correlation analysis to detect relationships between bacterial taxa and numerical clinical and anthropometrical metadata. The results obtained for the normal weight group, revealed that phylum Firmicutes had members like the genus *RFN20* which negative correlation with age; the genus *Anaerostipes* had negative correlation with HDL-cholesterol and positive with waist/hip ratio; the genus *Phascolarctobacterium* had negative correlation with HDL-cholesterol, and positive with LDL-cholesterol/HDL-cholesterol ratio and Cholesterol/HDL-cholesterol ratio; The family *Peptostreptococcaceae* had negative correlation with age; the genus *Enterococcus* had a negative correlation with waist/hip ratio, and the family *Ruminococcaceae* had a positive correlation with waist/hip ratio. In the case of the phylum *Bacteroidetes*, the family *Rikenellaceae* had negative correlation with HDL-cholesterol; the genus *Odoribacter* had negative correlation with HDL-cholesterol, age, and positive correlation with LDL-cholesterol/HDL-cholesterol ratio; the genus *Prevotella* had negative correlation with waist/hip ratio, and the family *Barnesiellaceae* had a positive correlation with triglycerides. Three less abundant phyla had interesting members, the phylum *Proteobacteria* with the family *Enterobacteriaceae* had a negative correlation with waist/hip ratio; the phylum *Actinobacteria* had the family *Coriobacteriaceae* with positive correlation with LDL-cholesterol, LDL-cholesterol/HDL-cholesterol, and cholesterol/HDL-cholesterol ratios; and the phylum *Actinobacteria* with the genus *Slackia* had a negative correlation with fat percentage, and a positive correlation with age (Figure 6A).

On the other hand, the results obtained for the obesity group revealed interesting correlations among fecal microbiota and the studied parameters. For instance, the phylum Firmicutes, whose relative abundance did not change after one month of dietary intervention including nopal, had several correlations. The family *Veilloneaceae* had positive correlation with LDL-cholesterol and cholesterol; the family *Ruminococcaeae* had negative correlation with triglycerides; the genus *Anaerostipes* had negative correlation with HDL-cholesterol; the genus *Blautia* had a positive correlation with LDL-cholesterol; the genus *Granulicatella* had positive correlation with waist/hip ratio; the genus *Eubacterium* had a positive correlation with glucose; the genus *Staphylococcus* had a negative correlation with cholesterol/HDL-cholesterol, LDL-cholesterol/HDL-cholesterol ratios, LDL-cholesterol and cholesterol; the genus *Streptococcus* had a negative correlation with LDL-cholesterol and cholesterol; the genus *Dialister* had a negative correlation with LDL-cholesterol/HDL-cholesterol ratio, and finally the genus *Dehalobacterium* had positive correlation with weight and BMI. The phylum Bacteroidetes had two positive correlations of the genus *Odoribacter*, with weight and fat percentage. Correlations among four less abundant phyla in the fecal microbiota of the obesity group participants were found: The phylum Proteobacteria had the genus *Rubellimicrobium* positively correlating with glucose, and the genus *Haemophilus* negatively correlating with LDL-cholesterol, and cholesterol. The phylum Actinobacteria had the genus *Colinsella* with a positive correlation with weight; the phylum Cyanobacteria had the family *Pseudanabaenaceae* positively correlating with the same parameter and the genus *Akkermansia*, of the phylum Verrucomicrobia, correlated negatively with HDL-cholesterol and positively with cholesterol/HDL-cholesterol ratio (Figure 6B).

We also performed PICRUSt2 analyses to find predicted metabolic pathways of relevance among the members of the fecal microbiota during the dietary intervention with nopal; however, no metabolic pathway with statistical significance was reported by the analysis.

## 4. Discussion

The gut microbiota is an important marker in health and disease, being altered in metabolic diseases such as obesity. The gut microbiota is prone to modification, especially by diet, which is one of the major influences on microbial signatures [1]. In the present work we used an energy restriction intervention adding nopal supplementation into the diets of a group of women with obesity, with the purpose of assessing the capacity of this food to improve the gut microbiota towards the intestinal microbiota of healthy people with normal weight. We observed the effect of dietary supplementation with nopal on the gut microbiota of women affected by obesity and normal weight women whose diets were supplemented with nopal under the same conditions for one month, identifying changes in bacterial taxa in both groups. At the same time, we observe bacterial associations with the biochemical and anthropometric markers evaluated.

In the individuals with obesity, *Parvimonas* was better represented at the beginning of the intervention, suggesting that caloric restriction, nopal supplementation and physical activity reduced the abundance of this bacterium. *Parvimonas* also increased in a study with obese Japanese people in comparison with lean people [21]. Moreover, in obese participants we did not find significant differences at phylum level after the dietary intervention supplemented with nopal, but we did find that *Prevotella*, *Roseburia*, *Lachnospiraceae* and *Clostridiaceae* showed an upward trend, while *Bacteroides*, *Blautia* and *Ruminococcus* (Appendix A), exhibited a tendency to decrease in the obesity group, but these trends were not significant. Since nopal is rich in dietary fiber [22], the results obtained in our study are consistent with works in which dietary fiber interventions were used [23,24]. In addition, these bacterial groups were highly reported as the indigenous microbiota of populations from rural communities, who have a long-term high-fiber consumption [25], as in a study performed in individuals from Papua New Guinea living a traditional lifestyle: these individuals exhibited a low abundance of *Faecalibacterium*, *Ruminococcus*, *Bifidobacterium*, *Bacteroides*, *Blautia* and high abundance of *Prevotella* [26]. Regarding the increasing tendency of *Prevotella* in the obese group after a month with nopal supplementation, an increase in this genus has also been reported in obese rats fed with a high fat and sucrose diet, supplemented with nopal for the same amount of time [10]. The reason could be that *Prevotella* species have been correlated with plant-rich diets abundant in carbohydrates and fibers [27]. *Prevotella* has the capacity to digest complex carbohydrates due to its genetic and enzymatic potential to break down cellulose and xylan from foods [27,28]. In addition, we found a negative correlation between *Prevotella* and waist/hip ratio in the normal weight group. In other studies, positive associations were found between *Prevotella* species and propionate production, which has essential roles in the prevention of weight gain by reducing decreasing hepatic lipogenesis [27,29].

In the normal weight group, we detected a reduction of *Bacteroides*, *Ruminococcus*, *Rikenellaceae*, and *Ruminococcaceae* and an increase of *Blautia*, *Eubacterium*, *Lachnospiraceae*, *Erysipelotrichaceae* and *Clostridiales*. After the nopal intervention, a similar tendency of reduction was observed in both groups for *Ruminococcus* and *Bacteroides,* and an increase of the *Lachnospiraceae* family. These tendences were also observed in multiple works relating to dietary interventions, reporting the *Lachnospiraceae* family as a well-known fiber-degrading bacterium [1] and a diet-responsive bacterium [30]. The decrease in *Bacteroides* could be attributed to a higher intake of fructans as result of nopal supplementation. It is known that some fructans could reduce levels of *Bacteroides*, as described in a double-blind intervention study with inulin-type fructans in Belgian women [31]. Other bacteria, such as *Roseburia* and *Eubacterium*, were enriched after the nopal intervention, these microorganisms are known as saccharolytic microbes highly responsive to changes in fiber intake [32]. In addition, they were reported as bacteria correlated with plant-based diets, inulin degraders [30]. It is important to consider that substantial variations in individual microbiota could influence dietary responsiveness [30].

In our study we observed that blood levels of glucose, total cholesterol, and HDL- cholesterol significantly decreased; this could be attributed to nopal’s multiple components, which contribute to changes in specific biochemical parameters as previously reported [22]. For example, the low glycemic index of nopal, could regulate blood glucose and serum insulin concentrations by modulating Glucose-Dependent Insulinotropic Polypeptide (GIP) levels [33], and the pulp pectin has showed anti-hyperlipidemic effects, reducing lipid absorption and increasing fecal sterol excretion, thus disrupting the enterohepatic circle [22]. At the same time the hypocholesterolemic action of prickly pear pectin was proved in non-diabetic, non-obese males [34]. On the other hand, similar results to our work were found in obese rats with nopal consumption [10], in which the addition of nopal in the diet promoted the decrease in total cholesterol, serum glucose and triglycerides. The authors attributed this finding to the presence of polyphenols, and soluble and insoluble fiber that change gut microbiota, since the incorporation of food with probiotic potential to the diet, can modify the gut microbiota composition, and as consequence lead to changes in host metabolism [10,35].

Obesity is characterized by chronic low-grade inflammation. The microbiota participates in regulating the metabolism through the production of secondary metabolites, such as the short chain fatty acid, in particular butyrate which has anti-inflammatory properties [35,36]. Many studies show that the dietary intake of fermentable carbohydrates can influence butyrate production stimulating the growth of butyrate-producing bacteria [35] In our work, *Roseburia* and *Eubacterium*, two major butyrate producers, showed an increased tendency of abundance after nopal intervention. In the case of *Eubacterium* we found a positive correlation with blood glucose levels, also reported by Larsen, in which the ratio of *Clostridium coccoides*/*Eubacterium rectale* positively correlated with plasma glucose concentration in diabetic participants [37]. Likewise, the oral treatment with *Eubacterium hallii* improved insulin sensitivity in severely insulin resistant *db/db* mice by increasing the butyrate production [38]. These results suggest that the gut microbiota might participate in the glucose metabolism.

For the lipidic metabolism, the *Coriobacteriaceae* family displayed strong correlation with lipidic parameters. Recent publications have shown gut-associated *Coriobacteriia* have positive effects in host lipid metabolism, improving gut health [39]. Concerning other bacterial correlations, the *Barnesiellaceae* family was negatively associated with body mass index, consistent with the data presented in a cohort of a normal weight and obese adolescences; the abundance of this family may be a microbial biomarker in healthy adolescents [40]. Regarding the groups included in this work, statistically significant differences were observed in all anthropometric and biochemical variables between the obesity group and the normal weight group, including age. This difference in age is directly related to the serious obesogenic problem that affects Mexicans, combined overweight and obesity affects three-quarters of the population over 20 years old. It is possible that the age difference could affect the composition of the intestinal microbiota due to the host–microbiota dynamic age-relationship; to avoid this confounder variable, we analyzed the groups separately. Our research group had already faced a similar situation when analyzing the intestinal microbiota of obese people and healthy people with normal weight in a previous published study [41]. On the other hand, the participants included in the present study belong to the same sociocultural and community group as the participants included in the previously referred-to study. In the current results we found bacteria whose abundance was associated with age only in the normal weight group; this could suggest that age influences gut microbiota, as reported by other authors [42]. Our normal weight group was not subjected to caloric restriction, since it was not ethically permitted by the Institution. It has been reported that caloric restriction causes limited changes in fecal microbial composition of non-obese adults [43]; consequently, we believe that diet modification should have changed the fecal microbiota diversity in the normal weight women. Finally, even when we do not observe significant differences in the bacterial diversity after the nopal supplementation, we observe trends that might suggest changes in the bacterial community. We suppose this effect could be more noticeable after increasing the intervention time.

## 5. Conclusions

The results presented in this work suggest that addition of fiber-rich nopal to the diet, caloric restriction, and physical activity, promote the development of certain bacteria in gut microbiota, modifying its composition in obese women, leading to changes in host metabolism, as suggested by the correlation between some bacterial species with biochemical and anthropometrical parameters. Moreover, nopal intervention induced changes in gut microbiota associated with biochemical and anthropometric parameters in normal weight women.

## Figures and Tables

**Figure 1 nutrients-14-01008-f001:**
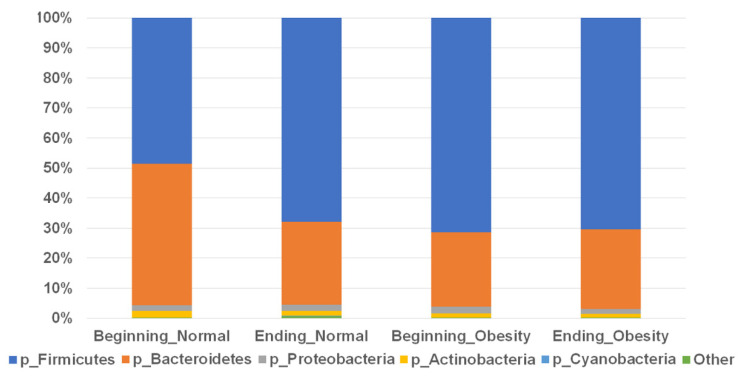
Relative abundance of bacterial phyla in fecal samples during nopal diet intervention. Data for normal weight and obese at beginning and ending of nopal diet. Sectors in bar plots indicate main phyla as shown by tag colors at the right side of the figure. Abundances of each bacterial phylum are shown as percentage in the Y-axis, and the groups names in the X-axis. “Other” include phyla with <0.50% relative abundance (Fusobacteria, Tenericutes, Verrucomicrobia, Chlorobi, Synergistetes, TM7, Thermi, Lentisphaerae, Chloroflexi, Acidobacteria, Fibrobacteres, Nitrospirae, Spirochaetes)—see Appendix A.

**Figure 2 nutrients-14-01008-f002:**
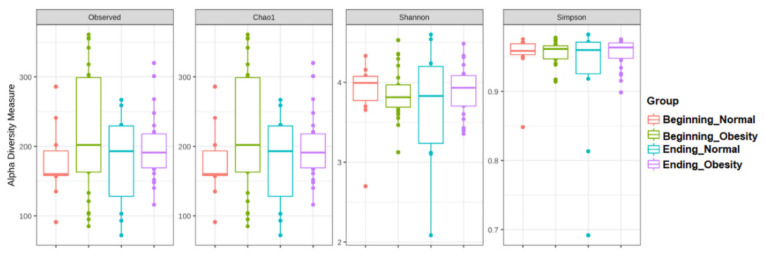
Alpha diversity of bacteria in fecal samples during nopal diet intervention. The figure shows data for normal weight and obesity groups at beginning and ending of nopal diet. The Y-axes indicate the values for the corresponding indexes: Observed number of species (Observed), expected bacterial richness (Chao1), and Shannon and Simpson diversity indexes. See Appendix A.

**Figure 3 nutrients-14-01008-f003:**
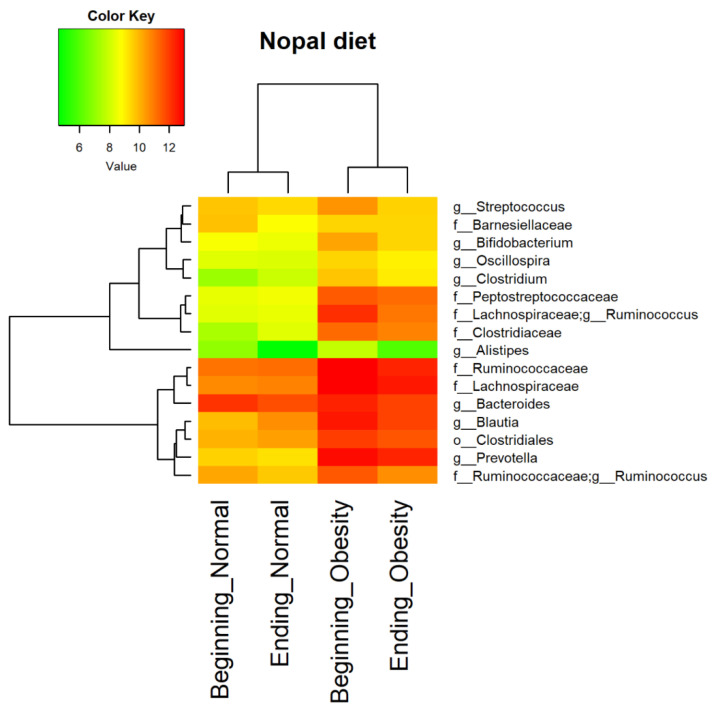
Bacteria with high relative abundance during nopal diet intervention. Differential abundance of the top 16 bacteria in fecal samples. The heat map shows differences among beginning and ending with nopal diet intervention on normal weight and obesity groups. Columns show the abundance of the top 16 bacterial taxa present in at least 95% of samples. Rows show the 16 bacteria clustered by its abundance in the group. The color key from green to red indicates differences in the natural logarithm of the absolute abundances obtained from the taxa-bar-plots.

**Figure 4 nutrients-14-01008-f004:**
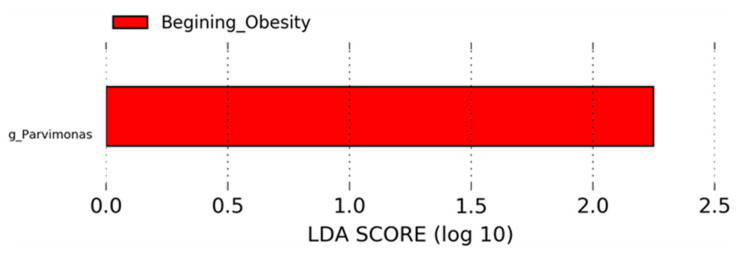
LEfSe analysis for obesity group during nopal diet intervention indicates *Parvimonas* genus is different in beginning obesity. Horizontal bars represent the effect size for each taxon. The length of the bars represents the log10 transformed LDA score, indicated by vertical dotted lines. The threshold on the logarithmic LDA score for discriminative features was set to 2.0. The name of bacteria with statistically significant changes in relative abundance is written alongside the horizontal lines. Taxon name is abbreviated as “g” for genus.

**Figure 5 nutrients-14-01008-f005:**
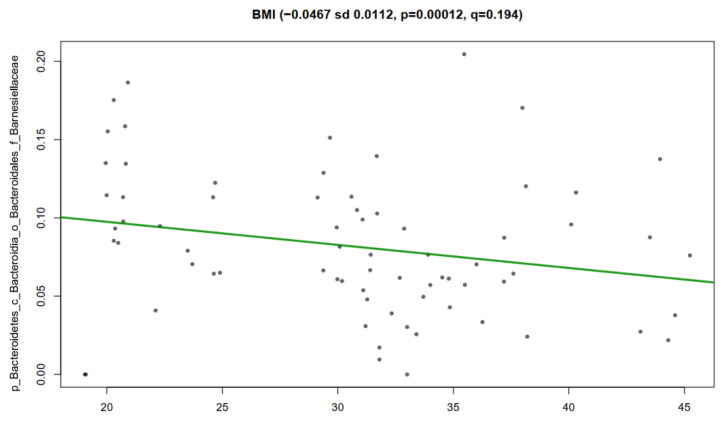
Multivariate linear associations of clinical metadata and bacterial relative abundance in all participants. Scatter plot explains the significant association of Body Mass Index (BMI) with the family *Barnesillaceae* as described in Materials and Methods. *y*-axes show the relative abundance of the bacteria; *x*-axes show the BMI data. Numerical data on top of graphic are coefficient (negative coefficient shows negative association between BMI and *Barnesillaceae*), sd—standard deviation; *p*-values, and FDR corrected *q*-values which are assigned by MaAsLin (v0.0.4). Obesity (BMI 29–45) and normal weight (BMI 19–24).

**Figure 6 nutrients-14-01008-f006:**
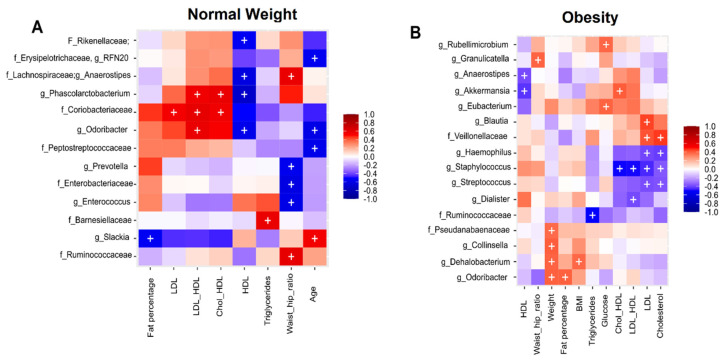
Spearman correlations of clinical metadata and bacterial abundance during nopal diet intervention. (**A**) Correlation in the normal weight group (**B**) Correlation in the obesity group. The heat map shows correlation between bacterial taxa and numerical metadata. Columns show the numerical metadata. Rows show the bacterial taxa. The color key from blue to red indicates the correlation value between −1 to +1; −1 indicates negative correlation and +1 positive correlation. The symbol “+” denotes a significance of *p* < 0.05.

**Table 1 nutrients-14-01008-t001:** Anthropometric and biochemical variables shown by participants.

	Normal Weight Group			Obesity Group		
Variable	Beginning *n* = 11	End *n* =11	*p*	Beginning *n* = 25	End *n* = 25	*p*
Age	22.1 ± 2.6 21 (18–27)	22.1 ± 2.6 21 (18–27)	ND	40.6 ± 10.7 42 (22–59)	40.6 ± 10.7 42 (22–59)	ND
Weight	54.3 ± 6.7 53 (45–65)	54.1 ± 6.7 53 (45–65)	0.79 ^b^	**84.9 ± 13.6** **78 (65–111)**	**83.1 ± 13.9** **76 (64–110)**	<0.001 ^a^
BMI	21.5 ± 1.9 20 (19–24)	21.4 ± 1.9 20 (19–24)	0.793 ^b^	**35.1 ± 4.5** **33 (30–44)**	**34.1 ± 4.7** **32 (29–45)**	<0.001 ^a^
Hip	70.9 ± 6.1 72 (62–85)	70.4 ± 6.1 71 (62–24)	0.974 ^b^	**99.7 ± 8.8** **99 (82–118)**	**99.1 ± 8.7** **99 (82–118)**	<0.001 ^b^
Waist	91.7 ± 8.6 91 (72–102)	91.6 ± 8.4 91 (72–102)	0.742 ^b^	114 ± 10 113 (100–139)	114 ± 10 113 (100–138)	0.380 ^b^
Hip/Waist	0.77 ± 0.05 0.7 (0.6–0.8)	0.77 ± 0.06 0.7 (0.6–0.7)	ND	**0.87 ± 0.04** **0.87 (0.8–0.9)**	**0.86 ± 0.04** **0.87 (0.8–0.9)**	0.008 ^a^
% Fat	26.4 ± 5.0 24 (20–36)	26.1 ± 4.9 25 (20–36)	0.887 ^a^	46.7 ± 5.3 47 (36–55)	46.1 ± 6.7 46 (28–59)	0.273 ^b^
Glucose	83.8 ± 4.5 85 (76–92)	82.5 ± 4.8 82 (73–90)	0.535 ^a^	**112 ± 41** **95 (82–248)**	**98 ± 26** **97 (81–150)**	0.030 ^b^
Tryglicerides	111 ± 24 170 (133–193)	106 ± 25 169 (115–188)	0.618 ^a^	172 ± 97 163 (50–493)	138 ± 42 133 (75–203)	0.069 ^b^
Cholesterol	167 ± 19 14 (9–25)	159 ± 24 14 (9–25)	0.390 ^a^	**190 ± 34** **192 (113–273)**	**178 ± 23** **181 (113–216)**	0.032 ^a^
HDL-Chol	60.5 ± 10.2 62 (40–78)	61.8 ± 10.3 63 (43–79)	0.775 ^a^	**41.3 ± 9.9** **40 (25–70)**	**39.2 ± 9.0** **37 (25–59)**	0.036 ^a^
LDL-Chol	84.9 ± 21.7 85 (44–117)	76.4 ± 26.6 88 (39–118)	0.420 ^a^	115 ± 30 120 (55–186)	111 ± 20 114 (59–137)	0.237 ^a^
LDL/HDL	1.47 ± 0.59 1.3 (0.5–2.9)	1.24 ± 0.49 1.3 (0.4–2.4)	0.338 ^a^	2.96 ± 1.13 2.7 (1.1–5.2)	2.99 ± 0.86 2.8 (1.0–4.5)	0.270 ^b^
Chol/HDL	2.85 ± 0.63 2.7 (1.8–4.4)	2.64 ± 0.56 2.6 (1.7–3.4)	0.415 ^a^	4.90 ± 1.55 4.6 (2.3–8.1)	4.76 ± 1.14 4.7 (2.2–6.9)	0.590 ^b^

Data shown as mean ± SD; below: median (minimum–maximum). ^a^ T-paired test was used to compare groups. ^b^ Wilcoxon Signed Rank Test was used to compare groups. ND = No Determined.

## Data Availability

The sequence FASTQ files, and the corresponding mapping file for all samples used in this study, were deposited in the NCBI BioProject ID PRJNA783637 Link https://www.ncbi.nlm.nih.gov/bioproject/PRJNA783637 (accessed on 20 February 2022).

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
