# Peer review of "Physical and Dietary Intervention with Opuntia ficus-indica (Nopal) in Women with Obesity Improves Health Condition through Gut Microbiota Adjustment"

_nutrients, 2022, doi:10.3390/nu14051008_

Round 1
Reviewer 1 Report
Overall, this study proposes to evaluate the impact of dietary intervention with nopal in obese women. The study design presents some problems that need to be corrected in order to make the right interpretation of the data. It is highly suggested that grammar and spelling is revised by a native English person. Too many errors to mention them all (see below please).
Under Introduction section: More references supporting beneficial effect of nopal should be provided. Separate paragraph (line 54-57) should be together with paragraph (line 47-52), why separated?
Line 60, need to specify if the occludin gene expression or protein was increased? In which organ? Not clear what the reference wants to support. Need to provide this information.
What is meant by nutritional maneuver on line 114?
How can the authors measure the effect of physical intervention (brisk pace for 30 mins per day ) and differentiate this effect from nopal dietary intervention?
How does the fiber content of nopal was measured? Insoluble/soluble total fiber?
For bacterial DNA analysis, why the V3 region instead of v3-V4 or V4 was used for the analysis? Authors need to clarify how the DNA quality was determined after electrophoretic fractionation. Is there a quality score calculated by the software (as usually is the case)? Did a numerical ratio was used to determine quality objectively? QSI ?
For core microbiota analysis, Line 192 what is “the near-table qza from demux qza and qiime taxa collapse”. The demux file generated by qiime shows the quality of the reads after trimming and it is not clear how this relates to a genera table and a collapse? This is very confusing.
For the multivariate association analysis why q-value <0.25 were considered significant. Need to explain the rationale for selecting this threshold.
For the statistical analyses did the authors considered the effect of repeated measures as the observations were collected from same subjects at different times (beginning and end of intervention)?
For the data used for Lefse analysis, typically the raw data is used so the program does the normalization and LDA calculations. It is not clear why data was run with a p.adjust () function in R before running Lefse? Have not seen this before. This section is very confusing and the justification for using several programs to analyze the same dataset set needs to be clarified. STAMP, Lefse and R use separate methods.
By looking at the Table 1, it is very obvious that there are significant differences among groups starting at the baseline. Authors did acknowledge this bias throughout the results (Diversity analysis and discussion), however the comparisons were still presented. It would make more sense to present the data separately within group with similar age. Does the nopal dietary intervention impacted the microbiome of obese subjects after a 4-week intervention? You should do this analysis separately. What specific taxa was significantly affected and how it correlated with changes in clinical parameters? The control group is not really a control group.
When differences in abundance of bacteria are presented it is very confusing. Are the authors measuring absolute abundance (as mentioned line 305) or relative abundance as mentioned in figure legend (line 310)? Figure 3 needs to have some statistic associated with it. Top 16 bacteria are shown. Based on the color key all of them are increasing with treatment in both groups, is this change significant? Supplementary table does not show any significant difference. Not sure why it is presented. What about any reductions in bacterial abundance? This type of comparison should be done separately with Lefse so taxa that drives the effect may be identified.
For the correlation analysis on Figure 5, did an adjustment for age was done given this is a very important covariate in this model?
Description of Figure 6 is very confusing. What is being measured? Association? (line 359) Correlation (line 360)? What is relation? The Legend for this figure does not explains what the “+ “sign denotes. Presumably significance?
In the discussion the level of propionate is not clear if is positive or negatively associated with Prevotella in this study. Was it measured? (line 401). The rationale to be included in discussion?
Some grammatical error or typos (examples):
“addition of rich in fiber nopal to the diet” (line 25)
‘the nopal also it has been used in traditional” (line 50)
“This group____no received any..” (verb) line 114
“as was described before, for their consumption” (line 114).
“analyzes” line 123
“star” line 143
“received” line 129
“Line 192 does not make sense”
“being more represented” line 281, Does this refer to abundance instead? The same for “were present with more abundance” (line 283).
Line 287 : data ___not showed (verb)
Line 289: no instead of not
Line 296: and____not showed (verb)
Line 297: beta diversity ____not showed (verb)
Paragraph describing Lefse graph (line 316) difficult to follow needs improvement
Line 354: inversely correlated instead?
Typically, bacterial families are italized.
When correlation are presented a statistic is required? Not provided in paragraph starting line 356.
In the discussion there is mention of reduction of Bacteroides, Ruminococcus etc… in control group. Where is this data?
Author Response
Manuscript ID: nutrients-1549459
Type of manuscript: Article
Title: Physical and dietary intervention with Opuntia ficus-indica (nopal) in women with obesity improves health condition through gut microbiota adjustment
Authors: Karina Corona-Cervantes, Alicia Parra-Carriedo, Fernando Hernández-Quiroz, Noemí Martínez-Castro, Juan Manuel Vélez-Ixta, Diana Guajardo-López, Jaime García-Mena, César Hernández-Guerrero.
Reviewer 1 (Round 1)
Comments and Suggestions for Authors
Overall, this study proposes to evaluate the impact of dietary intervention with nopal in obese women. The study design presents some problems that need to be corrected in order to make the right interpretation of the data.
Introduction section: (4)
1- It is highly suggested that grammar and spelling is revised by a native English person. Too many errors to mention them all (see below please).
Answer: we thank the reviewer for the advice, the English language grammar and use have been thoroughly reviewed and corrected along the text.
2- More references supporting beneficial effect of nopal should be provided.
Answer: we thank the reviewer for this suggestion that allows us to improve our manuscript. New references have been provided, and added to the paragraph, see line 54.
3- Separate paragraph (line 54-57) should be together with paragraph (line 47-52), why separated?
Answer: we thank the reviewer for the constructive remark, the paragraphs were arranged, see lines 51-51.
4- Line 60, need to specify if the occludin gene expression or protein was increased? In which organ? Not clear what the reference wants to support. Need to provide this information.
Answer: the sentence was modified and supplemented with pertinent information, as the reviewer suggested, see lines 61-62.
Methods Section (8)
5- What is meant by nutritional maneuver on line 114?
Answer: the reviewer made an opportune question. The paragraph of former line 114 indicates that the normal weight group did not receive any caloric restriction food plan during the supplementation with nopal during the 4 weeks. The paragraph has been changed as follows: “The group with normal weight did not receive any caloric restriction food plan during the nutritional supplementation with nopal during the 4 weeks” see lines 113-114.
6- How can the authors measure the effect of physical intervention (brisk pace for 30 mins per day ) and differentiate this effect from nopal dietary intervention?
Answer: this is another interesting question that we should clarify. In our study, it is not possible to separate the effect of nopal dietary intervention from the physical intervention. The study was designed to observe the effect of the sum of both interventions over the changes in clinical, biochemical, anthropometric, and intestinal microbiota markers in the participants. In any nutritional intervention aimed to reducing body weight in a person with obesity, implementing or increasing a physical activity program is always included. In our study, moderate physical activity was established for the participants. More intense physical activity many times is cause of no observance or desertion of the nutritional intervention.
7- How does the fiber content of nopal was measured? Insoluble/soluble total fiber?
Answer: in our study, the total fiber content of the nopal was not determined. A standard total dietary fiber content value of 2.2 g/100 g was used based on previously reported studies (Shetty et al., 2012; López-Romero et al., 2014; Ventura-Aguilar et al., 2017). However, to avoid as much as possible variations in the nutritional content of the used nopal in the study, the produce was purchased from same distributor who obtained them from the same zone and producer. This information is described in the material and methods section.
8- For bacterial DNA analysis, why the V3 region instead of v3-V4 or V4 was used for the analysis?
Answer: we appreciate the reviewer’s question on this matter. In published work from our laboratory (García-González et al., 2022; Sánchez-Salguero et al., 2021; Villalobos-Flores et al., 2021; Bello-Medina et al., 2021; Hernández-Quiroz et al, 2021; Cuervo-Zanatta, et al., 2021), we have successfully used the V3 polymorphic region of the 16S rRNA gene for sequencing using PGM Ion Torrent equipment and technology. It is important to mention that although recent published systematic work on bacterial diversity characterization based on the 16S rRNA gene variable regions, indicates that V4 is the most prominent V region for achieving good domain specificity, higher coverage, and broader spectrum in the bacteria domain; same work also states that single V3 and V6 regions are also reliable and comparable for the overall coverage from phylum to genus level (Zhang et al., 2018). In addition, a PubMed Advanced searching in “All fields” using “V3 16S rRNA” and “microbiota” as keywords produced 1,316 documents, while same search for “V4 16S rRNA” and “microbiota” produced 1,603 documents.
9- Authors need to clarify how the DNA quality was determined after electrophoretic fractionation. Is there a quality score calculated by the software (as usually is the case)? Did a numerical ratio was used to determine quality objectively? QSI ?
Answer: we thank the reviewer for getting this issue to our attention. The DNA was reviewed by electrophoretic fractionation in 0.5% agarose gels, while the level of purity was assessed by determination of the 260/ 280 ratio which were usually approximately 1.8.
10- For core microbiota analysis, Line 192 what is “the near-table qza from demux qza and qiime taxa collapse”. The demux file generated by qiime shows the quality of the reads after trimming and it is not clear how this relates to a genera table and a collapse? This is very confusing.
Answer: we apologize for the unclear writing and thank the reviewer for the opportunity to correct this issue in our text. The paragraph has a confusing redaction. We used the table.qza (feature table with quality of the reads after trimming) and the taxonomy.qza files (Taxonomic annotations for features, as genera) to obtain the abundance from the features (ASVs) using “qiime taxa collapse” program. The resulting files from this analysis are required to determine the core microbiota as input files. To avoid confusing information and misinterpretation, the paragraph was amended, see lines 185-187.
11- For the multivariate association analysis why q-value <0.25 were considered significant. Need to explain the rationale for selecting this threshold.
Answer: the microbiome Multivariable Association with Linear Models (MaAsLin) analysis we made, uses the max significance with a q-value threshold default <0.25 for significance (https://huttenhower.sph.harvard.edu/maaslin/).
12- For the statistical analyses did the authors considered the effect of repeated measures as the observations were collected from same subjects at different times (beginning and end of intervention)?
Answer: the reviewer raised and interesting issue. For every participant in each group, we made one measurement for each variable at the beginning and end of intervention. To perform the statistical analyzes in the normal weight and obesity groups, the effect of repeated measures was considered, for which the paired t-statistic test was used for data with parametric distribution, or Wilcoxon for non-parametric data.
13- For the data used for Lefse analysis, typically the raw data is used so the program does the normalization and LDA calculations. It is not clear why data was run with a p.adjust () function in R before running Lefse? Have not seen this before. This section is very confusing and the justification for using several programs to analyze the same dataset set needs to be clarified. STAMP, Lefse and R use separate methods.
Answer: we apologize again for the confusing narrative in the text. To be strict in our data analyses, we use the Benjamini-Hochberg correction assessment for all the p-values obtained from Lefse analysis (after the analysis). This correction was performed using the p.adjust () function in R and only for the Lefse Data. STAMP on the other hand, was used for the statistical analysis of the PICRUST data. We do not use several programs to analyze the same datasets. For a better understanding, the pertinent arranges were made to the text, see lines 220-223.
Results
14- By looking at the Table 1, it is very obvious that there are significant differences among groups starting at the baseline. Authors did acknowledge this bias throughout the results (Diversity analysis and discussion), however the comparisons were still presented. It would make more sense to present the data separately within group with similar age. Does the nopal dietary intervention impacted the microbiome of obese subjects after a 4-week intervention? You should do this analysis separately. What specific taxa was significantly affected and how it correlated with changes in clinical parameters? The control group is not really a control group.
Answer: we really appreciate the reviewer’s analysis of our data, and her/his clear insight to propose a better alternative way to present them. We took it into full consideration, and we have redefined the studied groups as follows: The group initially named “Control” is now called “Normal Weight” group, while the group initially named “Study”, is now called “Obesity” group. After the suggestion, data, analyses, and results are presented separately. On regard of the disparity observed in the age range of the groups, at population level, Mexico lives under epidemic conditions for overweight and obesity, where the prevalence of 38.4% for adolescents (12 to19 years-old) increases dramatically to 75.2% for adults (≥20 years-old) (Tamayo-Ortiz et al., 2021). This is one of the reasons why during sampling, most cases of Obesity are categorized to the 22–59 years-old range and are not abundantly present in the 18–27 years-old range, as seen in “Table 1”. In our study, monitored variables for instance weight, BMI, hip, hip/waist, glucose, cholesterol, and HDL-chol, are improved in the “Obesity” group, after four weeks of nopal dietary intervention, along with statistically significant changes in the abundance of important members of the fecal microbiota (e.g. Staphylococcus sp., Streptococcus sp., Bifidobacterium sp., etc…), which correlates with anthropometric and biochemical variables. It is important to mention that this occurs with no modification for the Alfa and Beta diversities parameters of the microbiota. Suitable modifications to the text, have been implemented.
15- When differences in abundance of bacteria are presented it is very confusing. Are the authors measuring absolute abundance (as mentioned line 305) or relative abundance as mentioned in figure legend (line 310)?
Answer: for core microbiota analyses we used the absolute abundance (heat map of Figure 3), in both cases it must be mentioned absolute abundance (line 300). This error was corrected.
16- Figure 3 needs to have some statistic associated with it. Top 16 bacteria are shown. Based on the color key all of them are increasing with treatment in both groups, is this change significant? Supplementary table does not show any significant difference. Not sure why it is presented. What about any reductions in bacterial abundance? This type of comparison should be done separately with Lefse so taxa that drives the effect may be identified.
Answer: we thank the reviewer for this observation. To make the core microbiota analyses as shown in the clustered heatmap (Figure 3), we used qiime feature-table core-features program to obtain the list and absolute abundance (calculated adding up counts from the OTU table), of bacteria present in at least 95% of the samples. The color and number of the key represents the absolute abundance of each featured bacteria, increasing in natural logarithm of counts from green to red. Green indicates the lowest while red the highest abundances. The clustered heatmap does not include any statistical test. This analysis was made along with the Linear discriminant analysis Effect Size (LEfSe) of Figure 4, to determine the fecal bacteria which most likely explain differences between the Beginning and End of the four weeks of nopal dietary intervention in each category.
17- For the correlation analysis on Figure 5, did an adjustment for age was done given this is a very important covariate in this model?
Answer: we made this analysis using all variables shown in Table 1 (including age). The multivariate linear association analysis of clinical metadata and the relative abundance of fecal bacteria in all participants shows statistically significant results for association of the family Barnesillaceae and BMI. It is clearly observed in Figure 5 that participants with a BMI of 29–45 which are in the Obesity group have lower relative abundance of members of the Barnesillaceae family. From this, it is assumed that a decrease in BMI after the four weeks of physical and dietary intervention with nopal, associates with an increase in the fecal Barnesillaceae family abundance (Figure 5).
18- Description of Figure 6 is very confusing. What is being measured? Association? (line 359) Correlation (line 360)? What is relation? The Legend for this figure does not explains what the “+ “sign denotes. Presumably significance?
Answer: again, the reviewer is right. We meant correlation on both cases since our Spearman correlation analysis measured the strength of the relationship between the variables. The term was changed to “correlation” in the original lines. In addition, the legend of Figure 6 was updated to better explain the results. Now it reads as follows: Spearman correlations of clinical metadata and bacterial abundance during nopal diet intervention. (A) Correlation in normal weight group (B) Correlation in obesity group. The heat map shows correlation between bacterial taxa and numerical metadata. Columns, show the numerical metadata. Rows show the bacterial taxa. The color Key from blue to red indicates the correlations value between -1 to +1; -1 indicates negative correlation and +1 positive correlation. The plus symbol “+” denotes a significance of p < 0.05.
Discussion
19- In the discussion the level of propionate is not clear if is positive or negatively associated with Prevotella in this study. Was it measured? (line 401). The rationale to be included in discussion?
Answer: we reported in our study the negative correlation of Prevotella and the waist-Hip index for normal weight group, here we suggest that this effect could be due to the association of Prevotella and higher levels of propionate, which has essential roles in the prevention of weight gain by reducing hepatic lipogenesis. Text was modified for a better understanding, lines 415-418.
20- Some grammatical error or typos (examples):
Answer: 20- All grammatical errors or typos were corrected during English language editing review.
21-Typically, bacterial families are italized.
Answer: we appreciate the reviewer’s remark. On regard of the Bacterial nomenclature, this issue is controversial since there appear to be rules according to different sources. For instance:
-CDC: Italics are used for bacterial and viral taxa at the level of family and below. All bacterial and many viral genes are italicized. https://wwwnc.cdc.gov/eid/page/scientific-nomenclature
-International Code of Nomenclature for algae, fungi, and plants: All taxon italicized in the code. International Code of Nomenclature (ICN) for Algae, Fungi and Plants (biocyclopedia.com)
-American Society of Microbiology: Names of all bacterial taxa (kingdoms, phyla, classes, orders, families, genera, species, and subspecies) are printed in italics. https://journals.asm.org/nomenclature
The reviewer suggestion was followed, and all bacterial family names were italicized in the text.
22-When correlation are presented a statistic is required? Not provided in paragraph starting line 356.
Answer: following the reviewer’s advice, the legend corresponding to Figure 6 was modified. In this legend it is mentioned that for the Spearman correlation analysis, there are positive and negative correlations with a statistical significance of p < 0.05.
23- In the discussion there is mention of reduction of Bacteroides, Ruminococcus etc… in control group. Where is this data?
Answer: we thank the reviewer for the opportunity to correct, the requested data are shown in the Supplementary Table 3.
References for Reviewer 1
Bello-Medina, P. C., Hernández-Quiroz, F., Pérez-Morales, M., González-Franco, D. A., Cruz-Pauseno, G., García-Mena, J., Díaz-Cintra, S., & Pacheco-López, G. (2021). Spatial Memory and Gut Microbiota Alterations Are Already Present in Early Adulthood in a Pre-clinical Transgenic Model of Alzheimer's Disease. Frontiers in neuroscience, 15, 595583. https://doi.org/10.3389/fnins.2021.595583
Cuervo-Zanatta, D., Garcia-Mena, J., & Perez-Cruz, C. (2021). Gut Microbiota Alterations and Cognitive Impairment Are Sexually Dissociated in a Transgenic Mice Model of Alzheimer's Disease. Journal of Alzheimer's disease : JAD, 82(s1), S195–S214. https://doi.org/10.3233/JAD-201367
García-González, I., Corona-Cervantes, K., Hernández-Quiroz, F., Villalobos-Flores, L. E., Galván-Rodríguez, F., Romano, M. C., Miranda-Brito, C., Piña-Escobedo, A., Borquez-Arreortúa, F. G., Rangel-Calvillo, M. N., & García-Mena, J. (2022). The Influence of Holder Pasteurization on the Diversity of the Human Milk Bacterial Microbiota Using High-Throughput DNA Sequencing. Journal of human lactation : official journal of International Lactation Consultant Association, 38(1), 118–130. https://doi.org/10.1177/08903344211011946
Hernández-Quiroz, F., Murugesan, S., Velazquez-Martínez, C., Villalobos-Flores, L. E., Maya-Lucas, O., Piña-Escobedo, A., García-González, I., Ocadiz-Delgado, R., Lambert, P. F., Gariglio, P., & García-Mena, J. (2021). The vaginal and fecal microbiota of a murine cervical carcinoma model under synergistic effect of 17β-Estradiol and E7 oncogene expression. Microbial pathogenesis, 152, 104763. https://doi.org/10.1016/j.micpath.2021.104763
López-Romero, P., Pichardo-Ontiveros, E., Avila-Nava, A., Vázquez-Manjarrez, N., Tovar, A. R., Pedraza-Chaverri, J., & Torres, N. (2014). The effect of nopal (Opuntia ficus indica) on postprandial blood glucose, incretins, and antioxidant activity in Mexican patients with type 2 diabetes after consumption of two different composition breakfasts. Journal of the Academy of Nutrition and Dietetics, 114(11), 1811–1818. https://doi.org/10.1016/j.jand.2014.06.352
Sánchez-Salguero, E., Corona-Cervantes, K., Guzmán-Aquino, H. A., de la Borbolla-Cruz, M. F., Contreras-Vargas, V., Piña-Escobedo, A., García-Mena, J., & Santos-Argumedo, L. (2021). Maternal IgA2 Recognizes Similar Fractions of Colostrum and Fecal Neonatal Microbiota. Frontiers in immunology, 12, 712130. https://doi.org/10.3389/fimmu.2021.712130
Shetty, A. A., Rana, M. K., & Preetham, S. P. (2012). Cactus: a medicinal food. Journal of food science and technology, 49(5), 530–536. https://doi.org/10.1007/s13197-011-0462-5
Tamayo-Ortiz, M., Téllez-Rojo, M. M., Rothenberg, S. J., Gutiérrez-Avila, I., Just, A. C., Kloog, I., Texcalac-Sangrador, J. L., Romero-Martinez, M., Bautista-Arredondo, L. F., Schwartz, J., Wright, R. O., & Riojas-Rodriguez, H. (2021). Exposure to PM2.5 and Obesity Prevalence in the Greater Mexico City Area. International journal of environmental research and public health, 18(5), 2301. https://doi.org/10.3390/ijerph18052301
Ventura-Aguilar, R. I., Bosquez-Molina, E., Bautista-Baños, S., & Rivera-Cabrera, F. (2017). Cactus stem (Opuntia ficus-indica Mill): anatomy, physiology and chemical composition with emphasis on its biofunctional properties. Journal of the science of food and agriculture, 97(15), 5065–5073. https://doi.org/10.1002/jsfa.8493
Villalobos-Flores, L. E., Espinosa-Torres, S. D., Hernández-Quiroz, F., Piña-Escobedo, A., Cruz-Narváez, Y., Velázquez-Escobar, F., Süssmuth, R., & García-Mena, J. (2021). The Bacterial and Fungal Microbiota of the Mexican Rubiaceae Family Medicinal Plant Bouvardia ternifolia. Microbial ecology, 10.1007/s00248-021-01871-z. Advance online publication. https://doi.org/10.1007/s00248-021-01871-z
Zhang, J., Ding, X., Guan, R., Zhu, C., Xu, C., Zhu, B., Zhang, H., Xiong, Z., Xue, Y., Tu, J., & Lu, Z. (2018). Evaluation of different 16S rRNA gene V regions for exploring bacterial diversity in a eutrophic freshwater lake. The Science of the total environment, 618, 1254–1267. https://doi.org/10.1016/j.scitotenv.2017.09.228
---end-of-text---

Reviewer 2 Report
Dear Authors,
Overall, the manuscript is well-written. Also, it was interesting to see how dietary intervention could alter the gut microbiota. However, there are a few questions that need to be answered to improve the manuscript.
- Figure S1 and S2 are missing in supplementary materials.
- Firmicutes to Bacteroidetes ratio changes depending on the age (infants, adults, and elderly individuals). In this context, age differences between control and study groups may be too significant to draw conclusions. Please justify the reason.
- Although no significant differences of Firmicutes and Bacteroidetes were observed between Control Beginning and Control End (Figure 1), often increase in Firmicutes to Bacteroidetes ratio is associated with obesity. This means when the study ended, people in the control group may gain more weight. But according to Table 1, the average weight of the beginning and end of the control group are 54.3 and 54.1 kg, respectively. Please provide the possible reason for this consequence.
Author Response
Manuscript ID: nutrients-1549459
Type of manuscript: Article
Title: Physical and dietary intervention with Opuntia ficus-indica (nopal) in women with obesity improves health condition through gut microbiota adjustment
Authors: Karina Corona-Cervantes, Alicia Parra-Carriedo, Fernando Hernández-Quiroz, Noemí Martínez-Castro, Juan Manuel Vélez-Ixta, Diana Guajardo-López, Jaime García-Mena, César Hernández-Guerrero.
Reviewer 2 (Round 1)
Comments and Suggestions for Authors
Dear Authors,
Overall, the manuscript is well-written. Also, it was interesting to see how dietary intervention could alter the gut microbiota. However, there are a few questions that need to be answered to improve the manuscript.
1- Figure S1 and S2 are missing in supplementary materials.
Answer: we thank the reviewer for the generous opportunity to correct this involuntary oversight. The requested figures were added to the supplementary material file.
2- Firmicutes to Bacteroidetes ratio changes depending on the age (infants, adults, and elderly individuals). In this context, age differences between control and study groups may be too significant to draw conclusions. Please justify the reason.
Answer: the reviewer’s observation is opportune. For a better interpretation of the data, we decided to analyze the two groups independently. The “Control” group is now called “normal weight”, and the “study group” is now called “obesity” group. After this change, data, analyses, and results are presented separately. This new approach avoids the age differences confounding effect.
3- Although no significant differences of Firmicutes and Bacteroidetes were observed between Control Beginning and Control End (Figure 1), often increase in Firmicutes to Bacteroidetes ratio is associated with obesity. This means when the study ended, people in the control group may gain more weight. But according to Table 1, the average weight of the beginning and end of the control group are 54.3 and 54.1 kg, respectively. Please provide the possible reason for this consequence.
Answer: the reviewer raised and interesting issue. Although Fig 1 seems to show a difference in percentage between beginning and end in the Normal weight group (formerly called control group), there is not a significant difference p = 0.168 (see data in Table S2). We think the short-term dietary intervention (1 month) was not long enough to produce significant change in the Firmicutes to Bacteroidetes ratio which could translate in weight change. In addition, the Figure 1 was corrected for a minor graphical error of data for Firmicutes abundance for the Normal weight group (formerly called control group).
---end-of-text---
